# Influence of Structure-modifying Agents in the Synthesis of Zr-doped SBA-15 Silica and Their Use as Catalysts in the Furfural Hydrogenation to Obtain High Value-added Products through the Meerwein-Ponndorf-Verley Reduction

**DOI:** 10.3390/ijms20040828

**Published:** 2019-02-14

**Authors:** Raquel López-Asensio, Carmen Pilar Jiménez Gómez, Cristina García Sancho, Ramón Moreno-Tost, Juan Antonio Cecilia, Pedro Maireles-Torres

**Affiliations:** Departamento de Química Inorgánica, Cristalografía y Mineralogía, Facultad de Ciencias, Campus de Teatinos, Universidad de Málaga, 29071 Málaga, Spain; lopezasensioraquel@gmail.com (R.L.-A.); carmenpjg@uma.es (C.P.J.G.); cristinags@uma.es (C.G.S.); rmtost@uma.es (R.M.-T.); maireles@uma.es (P.M.-T.)

**Keywords:** furfural, furfuryl alcohol, alkylfurfuryl ether, MPV reaction, hydrogen catalytic transfer, Zr-SBA-15, structure modifying agents

## Abstract

Zr-doped mesoporous silicas with different textural parameters have been synthesized in the presence of structure-modifying agents, and then characterized by X-ray diffraction (XRD), transmission electron microscopy (TEM), N_2_ adsorption-desorption at −196 °C, NH_3_ thermoprogrammed desorption (NH_3_–TPD), CO_2_ thermoprogrammed desorption (CO_2_–TPD), and X-ray photoelectron spectroscopy (XPS). These porous materials were evaluated in the furfural hydrogenation through the Meerwein-Ponndorf-Verley (MPV) reaction. The catalytic results indicate that the catalyst synthesized under hydrothermal conditions and adding a pore expander agent is more active and selective to furfuryl alcohol. However, the Zr-doped porous silica catalysts that were synthesized at room temperature, which possess narrow pore sizes, tend to form *i*-propyl furfuryl and difurfuryl ethers, coming from etherification between furfuryl alcohol (FOL) and isopropanol molecules (used as H-donor) by a S_N_2 mechanism.

## 1. Introduction

The depletion of fossil fuels has generated a need to look for alternative renewable resources to satisfy the world energetic demand. Several energy sources have been proposed and, among them, biomass is the only one that is able to provide both energy and chemicals, which are traditionally obtained from fossil fuels.

Biomass is a widely available raw material, which is distributed throughout the planet. This fact can favor the energy self-sufficiency of countries, the decline alongside speculation, and avoid political conflicts due to economic interests. The use of biomass to obtain energy and chemicals will be sustainable as long as biomass could not interfere with the food chain, since the consumption of edible biomass can cause a rise in prices and an increase in the economic differences between countries. This is the case of lignocellulosic biomass, with three major components: lignin, cellulose, and hemicellulose [1]. When focusing on the hemicellulose, this fraction is formed from C5 and C6 monosaccharides, mainly pentoses, such as xylose and arabinose, and even hexoses, such as galactose and mannose, which are linked by a variety of glycosidic bonds, in contrast to cellulose, which is a homopolymer of glucose that is linked with only β-1,4-glycosidic bonds [1,2]. These bonds can be hydrolyzed in their respective monomers by an acid treatment, which in turn can be subjected to a dehydration reaction to generate furfural from C5 carbohydrates [2,3].

Besides bioethanol, furfural (FUR) is the second most prolific chemical that is produced in the sugar platform of biorefineries (300,000 tons year^−1^) [3,4]. The great interest of FUR is attributed to its physicochemical properties, with applications as extractant for the removal of aromatics from oils for upgrading the relationship between viscosity and temperature, from diesel in order to improve ignition properties, for the cross-linking in polymers and as fungicide or nematocide [5]. Although FUR has a wide variety of applications, the greatest interest in this compound is attributed to its chemical composition (an aldehyde (CHO) and a furan system), making FUR a versatile molecule for a large spectrum of reactions. Thus, the aldehyde group of FUR can suffer reduction to alcohols, oxidation to carboxylic acids, decarbonylation, aldol condensations, acylation, acetylation, reductive amination to amines, or Grignard reactions, while the furanic ring can be subjected to hydrogenation, oxidation, nitration, halogenation, or opening reactions [2,5,6].

In the case of furfural hydrogenation to furfuryl alcohol (FOL), copper chromite is used to catalyze the process on an industrial scale. This product is considered as a valuable compound, since it is used for the production of resins and tetrafurfuryl alcohol, intermediate in fragrance production, as well as a precursor in the synthesis of lysine and vitamin C [6,7,8]. However, the use of copper chromite as commercial catalyst [9,10] in hydrogenation reactions is generating controversy, since chromium species can be toxic as Cr(VI). When considering this premise, much attention is being paid to the development of environmentally friendly catalysts. In this sense, several transition metals, such as Cu [11,12,13], Ni [12,14], Pd [12], Pt [15], or Ru [16] have shown an excellent catalytic performance to replace the traditional copper chromite, both in gas and liquid phase, in the presence of H_2_. However, the use of alcohols as hydrogen donors, through a catalytic transfer hydrogenation (CTH) reaction, is emerging as an alternative to H_2_ for reduction processes of aldehydes and ketones [17,18]. This process does not require high H_2_ pressures, as those that are employed in some reduction reactions, mainly in liquid phase [18].

In earliest studies, Meerwein, Ponndorf, and Verley (MPV) carried out studies independently in 1925–1926 [19,20,21], and they pointed out that a secondary alcohol could be used as sacrificing alcohol for hydrogen donation to reduce aldehydes or ketones, a process taking place on Lewis acid sites. In recent studies, it has been reported that Lewis acid sites promote hydrogen transfer from primary and, mainly, secondary alcohols to carbonyl groups through a six-membered intermediate [18,22]. Initially, homogeneous Lewis catalysts permitted to accomplish the MPV reaction (metal complexes and alkoxides) [23,24,25]. In spite of these homogeneous catalysts reached high yield values, they have associated cost overruns that are related to the separation and purification of the products. In addition, the catalytic activity of these homogeneous catalysts is strongly influenced by moisture. For this reason, solid heterogeneous catalysts have emerged as an alternative to homogeneous ones, since they can be easily recovered and regenerated. In this sense, metal oxides, such as Al_2_O_3_ [26,27,28], ZrO_2_ [28,29,30,31,32], or MgO [28,30,33,34], have shown activity in the reduction of aldehydes and ketones in obtaining their respective primary or secondary alcohol using a sacrificing alcohol. Other authors have reported that the incorporation of several M(IV) cations, such as Zr(IV) [22,35,36,37,38,39], Sn(IV) [22,35,36,40,41,42,43], or Hf(IV) [36,39], into a zeolitic framework generated a high density of Lewis acid sites, thus providing an extraordinary catalytic performance in the MPV reaction. It has also been demonstrated the synergetic effect in the MPV reaction of acid and basic sites coexisting in catalysts, such as Al_2_O_3_ [26], ZrO_2_ [31], or calcined hydrotalcites [44,45,46].

One of the main alternatives widely utilized in heterogeneous catalysis is the use of a porous material as support, with a high surface area, narrow pore distribution, and high thermal stability. In this sense, the best support that meets these parameters is mesoporous silica. In 1998, Zhao et al. synthesized, in the Santa Barbara University (USA), a mesoporous silica that was denoted as SBA-15 [47,48], which displayed uniform cylindrical mesopores with a diameter of 5–30 nm, hexagonally arranged, using amphiphilic triblock copolymer and oligomeric templates with poly (ethylene oxide) blocks as a pore directing agent [48]. The pore diameter of the channels can be modulated by the inclusion of pore expanders, like benzene-derived compounds [49,50]. In the same way, the channel length can be modified by the incorporation of fluoride species in the synthesis step, since these ions limit the polymerization of silicon species [49,51]. On the other hand, the acidity of the SBA-15 can be modulated by the insertion of heteroatoms, such as Al, Ti, or Zr, in its structure [52]. When considering that SBA-15 possesses high surface area and tunable porosity and ZrO_2_ species can provide both Lewis acid and basic sites [31], SBA-15 that is doped with Zr seems to be an appropriate candidate for the MPV reaction. Thus, Iglesias et al. evaluated the catalytic activity of the Zr-SBA-15 in the MPV reaction of FUR, obtaining FOL and alkyl furfuryl ethers as products [53]. In the same way, several authors have pointed out that microporous molecular sieves bearing isolated Zr sites are more effective catalysts in comparison to bulk zirconia [39,53].

The present research evaluates the influence of the width and the length of pores on the catalytic behavior in the MPV reaction of FUR to obtain valuable products. For this purpose, a pore expander agent has been incorporated in the synthesis step, as well as fluoride ions, in order to limit the polymerization of the silicon alkoxide in the formation of a SBA-15 silica doped with Zr, with a Si/Zr molar ratio of 5. The physico-chemical properties of these Zr-doped SBA-15 silica materials were evaluated and related with the catalytic behavior in the MPV of FUR.

## 2. Characterization of the Catalysts

The structural order of the Zr-doped porous silicas was evaluated by small-angle X-ray scattering (SAXS) (Figure 1). Thus, two defined bands can be observed: the first one being located between 2θ = 0.73–0.83° is ascribed to the *d*_100_ reflection, while that close to 2θ = 1.00° is attributed to the overlapping of the *d*_110_ and *d*_200_ reflections, which is in agreement with the data previously reported in the literature [54]. From SAXS data, it is also noteworthy that the incorporation of structure modifying agents in the synthesis step, both pore expander (trimethylbenzene, TMB) and fluoride species that decrease the length of the channels, causes a decrease in the intensity of these peaks. This fact suggests that the incorporation of these modifying agents leads to the formation of less ordered structures, as was previously demonstrated by other authors [50,54]. On the other hand, high-angle XRD patterns of the Zr-doped porous silicas do not evidence the phase segregation of ZrO_2_ species, indicating that the Zr species are forming part of silica walls and/or well dispersed on the internal and external silica surfaces [55].

The morphology of catalysts was evaluated from their TEM micrographs (Figure 2). It can be observed that the Zr-doped silicas aged at room temperature display a porous structure with narrower pores than those found in structures synthesized under hydrothermal conditions (Figure 2A). Among the materials synthesized at room temperature, the use of structure modifying agents and, in particular, fluoride ions, led to well defined pores with larger sizes (Figure 2B,C). On the other hand, the increase in aging temperature also generates these larger pores (Figure 2D), whereas the addition of structure-modifying agents gives rise to worse-defined structures (Figure 2E), including the formation of the so-called mesocellular foam (MCF) structures (Figure 2F) when both TMB and fluoride ions are added [49,51,56].

The microporosity [57] and the mesoporosity [58], as well as the pore size distribution [59] of the Zr-doped silicas have been determined from their N_2_ adsorption-desorption isotherms (Figure 3A and Table 1). According to the IUPAC classification, the adsorption isotherms of materials prepared at room temperature can be considered as type IV(a), characteristic of mesoporous adsorbents [60]. The existence of hysteresis loops indicates the capillary condensation of N_2_ molecules, which is directly related to the pore size, being appreciable when the pore is wider than 4 nm. In the case of samples obtained under hydrothermal conditions, the adsorption isotherms also are of Type IV(a), although they evolve towards Type II, which are typical of macroporous adsorbents, as suggests the clear increase in N_2_ adsorption at higher relative pressures [60]. With regard to the hysteresis loops, it is well known that SBA-15 is usually fitted as type H1, which is indicative of a narrow range of uniform mesopores. However, the incorporation of Zr species during the synthesis seems to modify the hysteresis loop, resembling the H2(b) type, which is associated to pore blocking, but the size distribution of neck widths is now much larger. This fact can be attributed to the different hydrolysis rate of silicon and zirconium alkoxides, which are used as precursors. Despite this, the XRD analysis did not indicate any phase segregation.

The evaluation of the microporosity by the *t*-plot method [56] (Table 1) reveals that the Zr-doped silica synthesized at room temperature displays a higher proportion of micropores, in comparison to those materials prepared under hydrothermal conditions. In this sense, it has previously been reported that the micelles of P-123, used as template, interacted with adjacent micelles at room temperature, leading to the typical parallel mesochannels of the SBA-15 structure, interconnected between them by microchannels, which facilitated the diffusion along the porous structure [61]. In the case of the Zr-doped silicas synthesized under hydrothermal conditions, the interaction between adjacent micelles diminishes. This fact implies a decrease of the microporosity of these hydrothermally prepared materials, as can be observed in Table 1. On the other hand, the use of structure-modifying agents hardly affects microporosity, so the aging temperature seems to be the key factor in order to tune the micropores fraction in these materials.

Similar to microporosity, BET surface area values follow the same trend, since materials that were aged at room temperature possess higher S_BET_ values than those prepared under hydrothermal conditions [58] (Table 1). On the other hand, the addition of structure-modifying agents under hydrothermal conditions increases macroporosity, as inferred from the amount of N_2_ adsorbed at higher relative pressures.

The estimation of the pore size distribution was done by the Density Functional Theory (DFT) method (Figure 3B) [59]. In the case of the Zr-doped silicas synthesized at room temperature, a broad pore distribution with pores between 1 and 12 nm is noticeable. These data seem to be in disagreement with the textural characteristics of mesoporous SBA-15 silica, which exhibits a narrow pore size distribution due to the formation of homogeneous cylindrical mesochannels. From these data, it is expected that the different hydrolysis rate of both alkoxides can lead to a porous less-defined structure with a broad pore size distribution. The incorporation of structure-modifying agents even provokes the formation of wider pore size distribution, with the maximum slightly shifted to higher pore size values. With regard to the Zr-doped silica synthesized under hydrothermal conditions, the pore size distribution is shifted to higher values, obtaining a maximum value that is close to 11 nm, which broadens and increases until 17 and 19 nm, after the addition of TMB or both TMB and fluoride ions, respectively.

The surface chemical composition was studied by XPS (Figure 4 and Table 2). The O 1*s* peak is observed about 532.5 eV, which can be ascribed to oxide species, whereas the photoelectron peaks in the Si 2*p* region (103.2 eV) and Zr 3*d* region (183.0 eV) are typical of silica and zirconia species, respectively [62]. The atomic concentrations (Table 2) reveal that, in all cases, the Si/Zr molar ratio varies between 34.1 and 36.7, so these values are far from the theoretical one (Si/Zr = 5). This fact could also be ascribed to the different hydrolysis rates of Si and Zr alkoxides. The XPS data suggests that Zr alkoxides suffer a faster hydrolysis rate than Si alkoxides, in such a way that a high proportion of Zr is incorporated into the inner regions of silica, thus decreasing the amount of surface Zr species. These data are in agreement with those previously reported in the literature, where the study of a Zr-doped mesoporous MCM-41 silica by XPS, after different times of Ar^+^ sputtering, allowed the observance of an enrichment of Zr from the outer to inner regions of this mesoporous material [63].

## 3. Catalytic Results

The catalytic behavior of this series of Zr-doped mesoporous silicas was studied in the furfural reduction by catalytic transfer hydrogenation, initially at a reaction temperature of 110 °C (Figure 5).

In Figure 5A, it can be observed that catalysts synthesized at room temperature display a similar behavior, since all of them reach similar FUR conversion (69–74%) after 6 h of reaction. Considering that the incorporation of structure modifying only slightly increases the pore size and pore volume, it seems clear that the modification of textural properties hardly affects the catalytic behavior for the catalysts prepared at room temperature.

However, in the case of the Zr-doped silicas synthesized under hydrothermal conditions, the differences are more pronounced. Thus, the HT catalyst reaches a FUR conversion of 78% after 5 h of TOS, slightly improving the catalytic performance with respect to catalysts obtained at room temperature. However, the expansion of the pore size by the incorporation of TMB (HT-TMB) causes an important improvement in the catalytic activity, attaining a maximum conversion of 92% after 6 h. This high FUR conversion could be ascribed to the easier diffusion of FUR and/or reaction products along its larger mesochannels, as previously evidenced from N_2_ adsorption-desorption isotherms. In addition, the use of higher pore width can limit the pore blockage, which would provoke the catalyst deactivation due to decay of the amount of available acid sites that are needed for the MPV reaction. On the contrary, the incorporation of fluoride species (HT-F-TMB), which favor the formation of shorter channels, led to the lowest conversion values, with only 61% after 6 h. These data suggest that the length of channels plays an important role in the FUR conversion, since longer channels could favor higher contact time between FUR molecules and active sites, resulting in higher FUR conversion values.

With regard to the selectivity (Figure 5B–D), it can be observed how the selectivity pattern is directly related to the textural properties. Thus, catalysts synthesized at room temperature are barely selective to FOL, alkylfurfuryl ethers being the main reaction products. Similarly to FUR conversion, catalysts prepared at room temperature display the same selectivity pattern, since the FOL yield is negligible (lower than 6%), while the *i*-propyl furfuryl ether (*i*-PFE) yield is in the range 43–51% and the difurfuryl ether (DFE) yield is between 17–22%. As was previously indicated, the FOL is a valuable product, which is used as monomer for the synthesis of furan resins, as well as in thermoset polymer matrix composites, cements, adhesives, coatings, and casting/foundry resins [6,7,8]. However, alkylfurfuryl ethers are also considered as interesting products, since they can be used as diesel additive due to their high cetane number [64,65]. In addition, DFE possesses potential in the food industry, because it is used as a flavoring ingredient to provide a coffee-like or nutty taste to food and beverages [66].

On the other hand, the catalytic study of materials prepared under hydrothermal conditions points out that the pore diameter modifies the selectivity pattern, since the FOL yield enhances as the pore size increases. Thus, the Zr-doped porous silica synthesized under hydrothermal conditions, HT, which displays higher pore width than the RT catalyst, provides an increase in the FOL yield (about 24% after 6 h); however, this catalyst still maintains a high yield of *i*-PFE (43%), while the DFE yield can be considered as negligible. The increase in the pore size by the incorporation of a pore expander produces a more pronounced modification of the selectivity pattern, since the HT-TMB catalyst reaches a maximum FOL yield of 67% after 6 h, whereas the *i*-PFE yield drops to 20%. However, the addition of fluoride species does not improve the selectivity towards none of the desired products, since the FOL and *i*-PFE yield are of 29% and 20%, respectively, after 6 h.

The catalytic data were compared with a bulk zirconia catalyst, synthesized by precipitation in basic medium, and subsequent calcination. It was found that the FUR conversion with the bulk ZrO_2_ catalyst is below most of the Zr-doped porous silicas. This fact can be explained by considering that the MPV reaction proceeds via Lewis acid sites, coming from Zr(IV) species, since Zr-doped mesoporous silica possesses a lower proportion of dispersed Zr(IV) species than bulk ZrO_2_. The main product obtained with bulk ZrO_2_ is FOL, although the presence of *i*-PFE and DFE is detected in minor proportions. In summary, it seems clear that the pore width and length of the mesochannels play a key role in the catalytic activity and, mainly, in determining the selectivity pattern. In this sense, smaller mesochannels favor the formation of *i*-PFE and DFE, which is in agreement with data reported in the literature, since furfural-derived ethers have been obtained by using solid acid catalysts with narrow pore size, as H-ZSM-5 zeolites [67,68].

After the catalytic test, the used catalysts were recovered by filtration and analyzed by several characterization techniques. Elemental analysis (CHN) showed that, in all cases, catalysts display a carbon content below 2.15 %, which is probably due to the strong adsorption of organic species. Among them, the catalysts with narrower pore size distribution, i.e. catalysts synthesized at room temperature, presented slightly higher carbon contents than those catalysts synthesized under hydrothermal conditions. In the same way, the catalysts with shorter channels, due to the incorporation of fluoride species, also showed a lower proportion of carbon. These data suggest that the presence of carbonaceous species can be associated to the adsorption of FUR and/or product in the channels of the Zr-doped porous silicas, which is also related with the rate of the diffusion along the channels. In the same way, surface atomic concentrations, as estimated from XPS data (Table 2), do not show the appearance of any new contribution in the C 1*s*, O 1*s*, Si 2*p,* or Zr 3*d* regions (Figure 4). However, it is noteworthy a slight increase of the C-concentration in all catalysts, which confirms the presence of organic deposits, although these are in a smaller proportion than those observed in gas-phase furfural hydrogenation [13]. On the other hand, it is also noticeable a slight decrease of the Si/Zr molar ratio after the MPV reaction. This fact could suggest that organic species tend to interact more strongly with silica in comparison to zirconia sites. This fact is striking, since Zr species are those that provide the acid and basic centers, so it could be expected the formation of hydrogen bonds between the silanol groups and the carbonyl group of FUR or hydroxyl group of FOL.

On the other hand, the textural properties of used catalyst were also evaluated. All of the catalysts suffer a decrease in the surface area, which could be ascribed to the loss of microporosity, as inferred from the *t*-plot data (Figure 6). This can be due to the deposition of FUR or reaction products in micropores. In addition, the catalyst with the highest FUR conversion and FOL yield (HT-TMB) is the catalyst with a lower proportion of micropores, so this catalyst should present less diffusional limitations when compared to the others.

In order to elucidate the pathways leading to the formation of alkyl-furfuryl ethers, FOL was fed instead of FUR, with the RT and HT-TMB catalysts, since these catalysts displayed a very different catalytic behavior (Figure 7). Feeding FOL, the conversion is around 88% for the RT catalyst, while the HT-TMB catalyst hardly reaches a conversion of 28%. Focusing on the *i*-PFE and DFE, these yields are slightly higher when the FOL is fed. This fact suggests that once FOL is formed, this is subsequently converted into the respective alkyl furfuryl ethers. This reaction proceeds on Lewis acid sites (Zr(IV) species) [37,38], which interact with the –OH group of the FOL to generate a better leaving group [53]. Afterwards a nucleophilic substitution (S_N_2) takes place with the sacrificing alcohol, although another FOL molecule can also act like nucleophile to form DFE. As was previously indicated, higher yields towards alkyl furfuryl ethers are attributed to the pore size, since catalysts with smaller pores lower the diffusion rate of FUR and/or FOL molecules along the channels, which favors the evolution of FOL towards alkyl furfuryl ethers.

The evaluation of acid properties by NH_3_–TPD is somewhat complicated, since all catalysts have a Si/Zr molar ratio of 5 and NH_3_ is a small molecule that is able to access to all pores independently of their sizes. The data obtained reveal that total acidity oscillates between 676 and 746 μmol g^−1^. From the obtained data, it can only be inferred that the number of acid sites is directly related to microporosity. However, it must be considered that some of these acid sites located in microporous could be inaccessible for FUR molecules, and can also be susceptible to suffer a faster deactivation by the formation of organic deposits, which cause the blockage of these acid sites. On the other hand, CO_2_–TPD analysis shows that Zr-doped porous silicas present a small proportion of basic centers, between 20–30 μmol g^−1^. This fact confirms the amphoteric character associated to the presence of Zr(IV) species into the siliceous framework.

Subsequently, the influence of reaction temperature on the catalytic performance was also evaluated for RT and HT-TMB catalysts (Figure 8). In both cases, FUR conversion directly increases with the temperature, from 58 up to 90%, raising the temperature from 90 to 130 °C, after 6 h, for the RT catalyst (Figure 8A). In the case of HT-TMB, FUR conversion is improved under similar experimental conditions from 67 to 99%. With regard to the reaction products, it has been previously indicated that the selectivity pattern is related to the length of the mesochannels. However, an increase in the reaction temperature also produces an evolution of the reaction products, since acid sites can catalyze the reaction between two alcohols to generate alkyl ethers and a water molecule as by-product. In this sense, the formation of *i*-PFE is favored in comparison to DFE, since *i*-PrOH is a more abundant alcohol in the reaction medium than FOL. In fact, Figure 8B clearly reveals that the FOL yield decreases for the HT-TMB catalyst after 2 h at 130 °C. This decrease is accompanied by the increase in the ether content, mainly for the *i*-PFE.

A key issue in heterogeneous catalysis is the catalyst reusing during several catalytic cycles (Figure 9). For this purpose, RT and HT-TMB catalysts were assayed for 2 h at 130 °C. Between each cycle, the solid was filtered, washed with the solvent, and dried overnight to be used in a next catalytic cycle under similar experimental conditions. The catalytic data reveal that both catalysts undergo a progressive deactivation as the number of reaction cycles increases. Thus, FUR conversion of the RT catalyst decreases from 63% after the first catalytic run to 37% after four cycles. However, the selectivity pattern hardly suffers modifications along several runs, since the main product is *i*-PFE, which directly decays with the FUR conversion from 50 to 33%. In the case of the HT-TMB catalyst, the FUR conversion decreases from 88% to 73% after the fourth cycle. This catalyst does not seem to show any drastic change in the product distribution, since a decrease in the formation of FOL is accompanied by a slight increase in the *i*-PFE yield. This fact could be ascribed to the partial occlusion of the channels by the presence of organic deposits, which can limit the diffusion rate along the channel, so the FOL can undergo an etherification reaction with the solvent, as was observed for the catalyst with a narrower pore size. Nevertheless, the deactivation by leaching of Zr(IV) species must be discarded, since the zirconium content that was analyzed in the reaction medium is below 0.003%. In all cases, the regeneration of the Zr-doped silica catalysts by calcination implies an increase of the FUR conversion, maintaining a similar selectivity pattern, although, in no case, the catalysts reached the conversion values obtained with the fresh catalysts, which was probably due to the calcination of the remaining organic matter being a exothermic process, which can affect the framework of the porous material.

As was indicated previously, the MPV reaction is promoted by an alcohol that transfers a hydride specie to the aldehyde of FUR. This process can be promoted by primary or secondary alcohols that act as sacrificing alcohol, although the primary alcohols can cause self-condensation reactions. On the other hand, the tertiary alcohols lack the available hydrogen in alpha position to promote MPV reaction [18,30]. Taking into account these considerations, *i*-propanol, 2-butanol, and cyclohexanol were evaluated as sacrificing alcohol in the MPV reaction, in the presence of RT and HT-TMB catalysts (Figure 10). In comparison with the use of *i*-propanol, 2-butanol slightly increases FUR conversion in both cases (Figure 10A). The data reported by other authors are in agreement with this fact [26,39], since a longer aliphatic chain decreases the polarity of the secondary alcohol, favoring the formation of a more stable six-membered intermediate with the Lewis acid sites, so that the transfer of the hydride species is easier. The use of cyclohexanol as sacrificing alcohol leads to poorest catalytic results, which is probably due to steric hindrance in the formation of six-membered intermediates. Moreover, the autogenous pressure generated by cyclohexanol is much lower than that obtained with 2-butanol and mainly *i*-propanol, as was indicated in a previous research [26], which can also limit the formation of products. With regard to the obtained products (Figure 10B–D), the nature of the sacrificing alcohol does not clearly influence the selectivity pattern, thus suggesting that the formation of alkyl furfuryl ethers only depends on the amount of Lewis acid sites, the reaction temperature, as well as the length and pore width of the channels.

Previous works have demonstrated that the coexistence of acid and basic sites can exert a synergistic effect on the catalytic behavior in the MPV reaction, as has been reported in the presence of ZrO_2_ [31,67], Al_2_O_3_ [26] or mixed oxides derived from hydrotalcites [46]. In this sense, pyridine or benzoic acid, which can interact with the acid and basic sites, respectively, were added into the reaction medium in equimolar amounts to the FUR, in order to evaluate the inhibition role on the acid and basic sites (Figure 11) of two catalysts (RT and HT-TMB) with different textural properties. From the catalytic data, it can be inferred that the incorporation of pyridine into the reaction medium barely affects the FUR conversion in the presence of both catalyst, while the addition of benzoic acid causes a strong decrease in the catalytic activity. These data could suggest that the MPV reaction is dominated by the basic sites of the Zr(IV) species, while the acid properties have a secondary role in the reaction. These data are in agreement with those that were obtained in the literature, since ZrO_2_-species dispersed on SBA-15 display a similar behavior in the MPV reaction of ethyl levulinate to γ-valerolactone, through the MPV reaction [69,70,71]. The incorporation of pyridine or benzoic acid does not affect the selectivity pattern in the case of the HT-TMB catalyst, since the main product is FOL. However, the RT catalyst modifies its selectivity pattern after partial blocking of active sites, mainly when benzoic acid blocks the basic sites. In this sense, the decrease of the amount of available basic sites retains the process in the MPV reaction, as it can be inferred from the increase in FOL yield, while the etherification reactions are partially inhibited. From these data, it could be concluded that the small proportion of basic sites must play an important role in the MPV reaction.

## 4. Materials and Methods

### 4.1. Reagents

The reagents that were used for the synthesis of the SBA-15 were: triblock copolymer Pluronic P123 (PEO_20_PPO_70_PEO_20_ average Mn ~5800, Sigma-Aldrich, Saint Louis, MO, USA) as template, tetraethylorthosilicate (TEOS 98%, Aldrich) as silicon source, zirconium propoxide (Zr(OCH_2_CH_2_CH_3_)_4_, 70 wt% in 1-propanol, Aldrich) as zirconium source, and hydrochloric acid (HCl 37%, VWR, Radnor, PA, USA). The pore expander was of 1,3,5-trimethylbenzene (TMB 98%, Aldrich), while the fluoride source was ammonium fluoride (NH_4_F 99.5%, Aldrich).

The chemicals used in the MPV reaction were: furfural (Sigma-Aldrich, 99%), 2-propanol (VWR, HPLC grade, 99.9%), 2-butanol (Sigma-Aldrich, 99.5%), and cyclohexanol (Sigma-Aldrich, 99%), used as sacrificing alcohols, and *o*-xylene (Sigma-Aldrich, 99.9%) was employed as the internal standard. The gases that were employed in this research were He (Air Liquide, Paris, France, 99.99%), H_2_ (Air Liquide, 99.999%), and N_2_ (Air Liquide, 99.9999%).

### 4.2. Catalysts Synthesis

By following the methodology described by Fulvio et al. with modifications, the synthesis of the mesoporous SBA-15 silica was performed [54]. Thus, Pluronic P-123 was dissolved in a solution of HCl (1.7 M) at 40 °C. Subsequently, both TEOS and Zr propoxide were mixed to obtain a Si/Zr molar ratio of 5 and they were added dropwise to the Pluronic solution. The molar composition of the obtained gel was 1 P123: 45.83 SiO_2_: 9.17 ZrO_2_: 350 HCl: 11100 H_2_O.

The synthesis of porous Zr-doped SBA-15 with pore expander was carried out according to the synthesis published by Santos et al. [50]. As was described above, Pluronic P-123 was dissolved in a solution of HCl (1.7M) at 40 °C and then the pore expander (TMB) was added, and after 30 min, the mixture TEOS and Zr-propoxide with a Si/Zr molar ratio of 5 was added dropwise. The final molar ratio in the synthesis gel was 1 P123: 45.83 SiO_2_: 9.17 ZrO_2_: 48 TMB: 350 HCl: 11100 H_2_O.

The incorporation of fluoride species in the Zr-doped mesoporous SBA-15 silica was carried out according to the procedure that was described by Vilarrasa et al., with modifications [55]. Pluronic P123 and NH_4_F were dissolved in 1.7 M HCl at 40 °C and then the pore expander (TMB) was added. After 30 min, the mixture TEOS and Zr-propoxide with a Si/Zr molar ratio of 5 was added dropwise. The final molar ratio in the synthesis gel was 1 P123: 45.83 SiO_2_: 9.17 ZrO_2_: 48 TMB: 350 HCl: 1.8 NH_4_F: 11100 H_2_O.

In all cases, after the incorporation of Si and Zr sources, some gels were aged at room temperature for 72 h, while others were aged at room temperature for 24 h, and then they transferred to a Teflon-lined autoclave of 90 mL, where it was hydrothermally treated at 80 °C for 48 h. After that, the obtained gels were filtered, washed with distilled water, and dried overnight at 60 °C. Finally, the template was removed by calcination at 550 °C with a heating rate of 1 °C min^−1^, for 6 h. Zr-doped porous silica were labeled, as indicated in Table 3.

The catalytic behavior of the Zr-doped porous silicas was compared with a bulk ZrO_2_. The synthesis of this ZrO_2_ was carried out from a solution of zirconium oxychloride, which was precipitated as hydroxide form by the dropwise addition of a NaOH solution. Subsequently, the obtained solid was dried overnight and calcined at 400 °C for 2 h, giving rise to ZrO_2_.

### 4.3. Characterization of the Catalysts

Small-angle X-ray scattering (SAXS) measurements were performed on a D8 DISCOVER-Bruker instrument (Billerica, MA, USA) at 40 kV and 40 mA. Powder patterns were recorded in capillary-transmission configuration by using a Göbel mirror (Cu Kα1 radiation) and the LYNXeye detector. The powder patterns were recorded between 0.2 and 10° in 2θ with a total measuring time of 120 min.

The morphology was studied by transmission electronic microscopy (TEM) using a Philips CM 100 Supertwin-DX4 microscope (Amsterdam, Netherlands). The samples were dispersed in ethanol and a drop of the suspension was put on a Cu grid (300 mesh).

The textural parameters were evaluated from the N_2_ adsorption-desorption isotherms at −196 °C, as determined by an automatic ASAP 2020 Micromeritics (Norcross, GA, USA). Prior to the measurements, the samples were outgassed overnight at 110 °C and 10^−4^ mbar. Micropore surface areas were obtained by de Boer’s *t*-plot method [57]. The specific surface area was determined by the Brunauer-Emmett-Teller equation (BET) using the adsorption data in the range of relative pressures, at which conditions of linearity and considerations regarding the method were fulfilled while taking into account that the N_2_ cross section is 16.2 Å^2^ [58]. The pore size distribution was determined from the desorption branch of the isotherm using the Nonlocal Density Functional Theory (NLDFT) [59]. The total pore volume was calculated from adsorbed N_2_ at *P*/*P*_0_ = 0.996.

X-ray photoelectron spectra were collected using a Physical Electronics PHI5700 (Chanhassen, MN, USA) spectrometer with non-monochromatic Mg Kα radiation (300 W, 15 kV, and 1253.6 eV) with a multichannel detector. Spectra of samples were recorded in the constant pass energy mode at 29.35 eV using a 720 μm diameter analysis area. Charge referencing was measured against adventitious carbon (C 1*s* at 284.8 eV). A PHI ACCESS ESCA-V6.0F software package was used for acquisition and data analysis. A Shirley-type background was subtracted from the signals. Recorded spectra were always fitted using Gaussian–Lorentzian curves in order more accurately to determine the binding energies of the different element core levels.

The amount of acid sites was determined by thermoprogrammed desorption of ammonia (NH_3_–TPD). Each experiment was carried out using 0.08 g of catalyst. In a first step, the catalyst was cleaned using a He flow (40 mL min^−1^) from room temperature to 550 °C, with a rate of 10 °C min^−1^ for 15 min, and it was then cooled under the same conditions until 100 °C. Later, the catalyst was saturated with pure NH_3_ for 5 min. Subsequently, a He flow (40 mL min^−1^) was passed to eliminate the physisorbed ammonia. Finally, temperature-programmed desorption was carried out by heating the samples from 100 to 550 °C at a heating rate of 10 °C min^−1^. The desorbed ammonia was quantified by a thermal conductivity detector (TCD). On the other hand, the amount of basic sites was quantified by the thermoprogrammed desorption of carbon dioxide (CO_2_–TPD). In each analysis, 300 mg of sample was pretreated under a He flow (40 mL min^−1^) at 550 °C for 15 min (10 °C min^−1^). The temperature was lowered to 100 °C and a pure CO_2_ stream (60 mL min^−1^) was subsequently introduced into the reactor for 30 min. The CO_2_–TPD was conducted between 100 and 550 °C under a helium flow (10 °C min^−1^ and 30 mL min^−1^) and the amount of CO_2_ evolved was analyzed using a TCD detector.

The leaching of the catalysts was determined by ICP-MS on Perkin Elmer (Waltham, MA, USA) spectrophotometer (NexION 300D). Previously, the samples were digested in an Anton Paar device (Multiwave 3000) by using HNO_3_, HCl, and HF.

### 4.4. Catalytic Tests

The MPV reaction of FUR was carried out in glass pressure reactors with thread bushing (Ace, 15 mL, pressure rated to 10 bars). In a typical experiment, 100 mg of catalyst were mixed with 100 mg of furfural dissolved in *i*-propanol, 2-butanol, and cyclohexanol, as sacrificing alcohol, maintaining an alcohol:furfural molar ratio of 50. Prior to each experiment, the reactors were always purged with helium. Reaction time was extended until 6 h, under continuous stirring (400 rpm), whereas the reaction temperature ranged between 90 and 130 °C. The temperature was controlled by a thermocouple directly in contact with a silicone bath. After the reaction time, the reactor was moved away from the silicone bath and then cooled in a water bath. Samples were microfiltered and analyzed by a gas chromatography (Shimadzu GC-14A) (Kioto, Japan), equipped with a Flame Ionization Detector and a CP-Wax 52 CB capillary column. The furfural conversion and selectivity were calculated, as follows:
$$\mathrm{Conversion}\;\left(\%\right)=\;\frac{\mathrm{mol}\;\mathrm{of}\;\mathrm{furfural}\;\mathrm{converted}}{\mathrm{mol}\;\mathrm{of}\;\mathrm{furfural}\;\mathrm{fed}}\;\times \;100$$
$$\mathrm{Selectivity}\;\left(\%\right)=\;\frac{\mathrm{mol}\;\mathrm{of}\;\mathrm{the}\;\mathrm{product}}{\mathrm{mol}\;\mathrm{of}\;\mathrm{furfural}\;\mathrm{converted}\;}\;\times \;100$$

## 5. Conclusions

Zr-doped porous silicas have been synthesized by the sol-gel method. In order to modify the textural properties, structure-modifying agents as TMB (to increase the pore volume) and fluoride ions (to reduce the length of channels) were added in the synthesis step. The surface chemical analysis showed that the Si/Zr molar ratio is much lower than the theoretical values due to the different hydrolysis rate of Si and Zr alkoxides.

Zr-doped porous silicas were evaluated in the catalytic transfer hydrogenation of FUR with alcohols. The catalytic results revealed that the textural properties are directly related to the FUR conversion, reaching the highest activity for the HT-TMB catalysts. In the same way, the selectivity pattern is also highly influenced by the pore size distribution and the size of channels. Thus, the catalyst synthesized under hydrothermal conditions, with a larger pore size, tends to form FOL as main product through the MPV reaction. However, the Zr-doped silicas prepared at room temperature, which display narrower pores, form *i*-PFE and DFE as products, with FOL in lower proportions. These data suggest that, in a first step, FOL is formed by the MPV reaction, and then FOL is etherified with another FOL molecule, or with the sacrificing alcohol that is present in the reaction medium, on acid sites.

The evaluation of several reaction parameters has revealed that an increase in reaction temperature improves the catalytic activity, although also modifies the selectivity pattern, mainly for the catalysts with the narrowest pore (RT catalyst), since the etherification reaction is favored at higher reaction temperatures. The catalytic data also pointed out that the type of secondary alcohol also plays an important role in the catalytic activity, since the use of a bulky alcohol, such as cyclohexanol, led to lower conversion values due to the steric effect, which disfavors the formation of the six-membered intermediate involved in the MPV reaction. The study of catalyst reuse has demonstrated that the catalysts suffer a progressive deactivation after each cycle, due to the formation of carbonaceous deposits that block the active sites. However, a calcination process has allowed reaching conversion values close to those obtained after the first run, so these catalysts can be regenerated. In this sense, the analysis of the Zr(IV) species in the reaction medium has ruled out the leaching of the Zr-doped silicas under the catalytic conditions.

Finally, the incorporation of a basic molecule as pyridine, or an acid molecule as benzoic acid, in the reaction medium showed that the blockage of basic sites has a strong influence on the catalytic behavior, while the blockage of acid sites has a lower influence on the catalytic activity. From these data, it can be inferred that the coexistence of acid and basic sites for the MPV reaction is necessary, with the basic sites being more susceptible to deactivation.

## Figures and Tables

**Figure 1 ijms-20-00828-f001:**
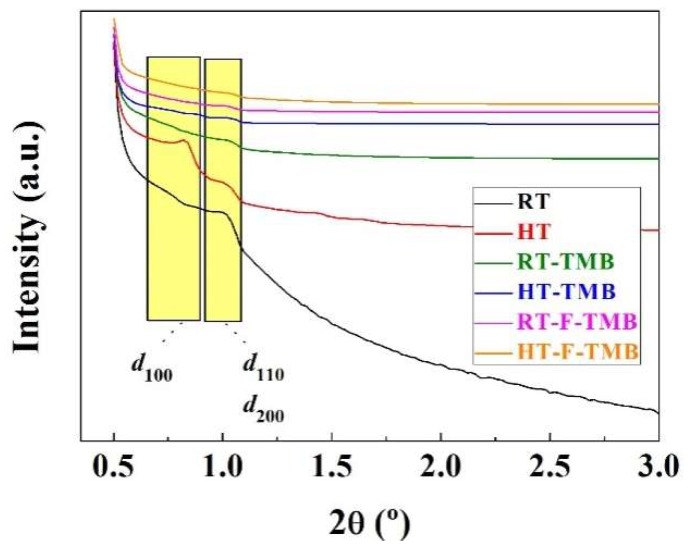
Small-angle X-ray scattering (SAXS) pattern of the Zr-doped silicas.

**Figure 2 ijms-20-00828-f002:**
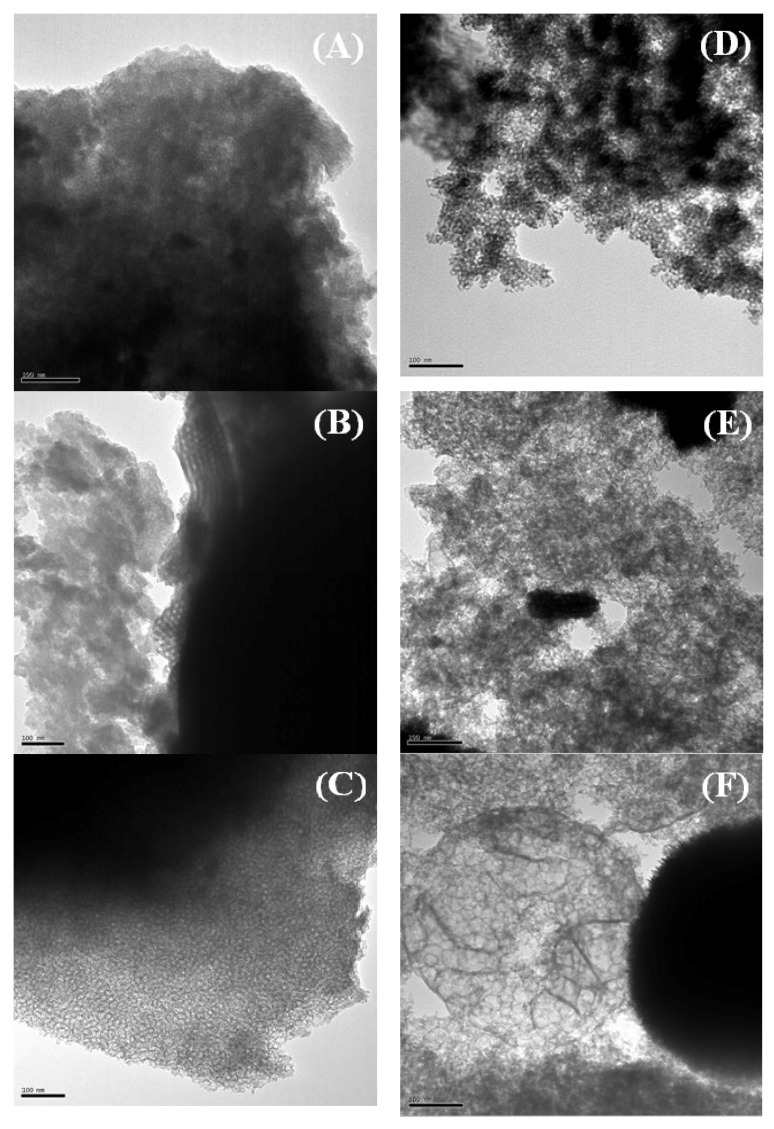
Transmission electronic microscopy (TEM) micrographs of RT (**A**), RT-TMB (**B**), RT-F-TMB (**C**), HT (**D**), HT-TMB (**E**), HT-F-TMB (**F**). Scale Bars: 100 nm.

**Figure 3 ijms-20-00828-f003:**
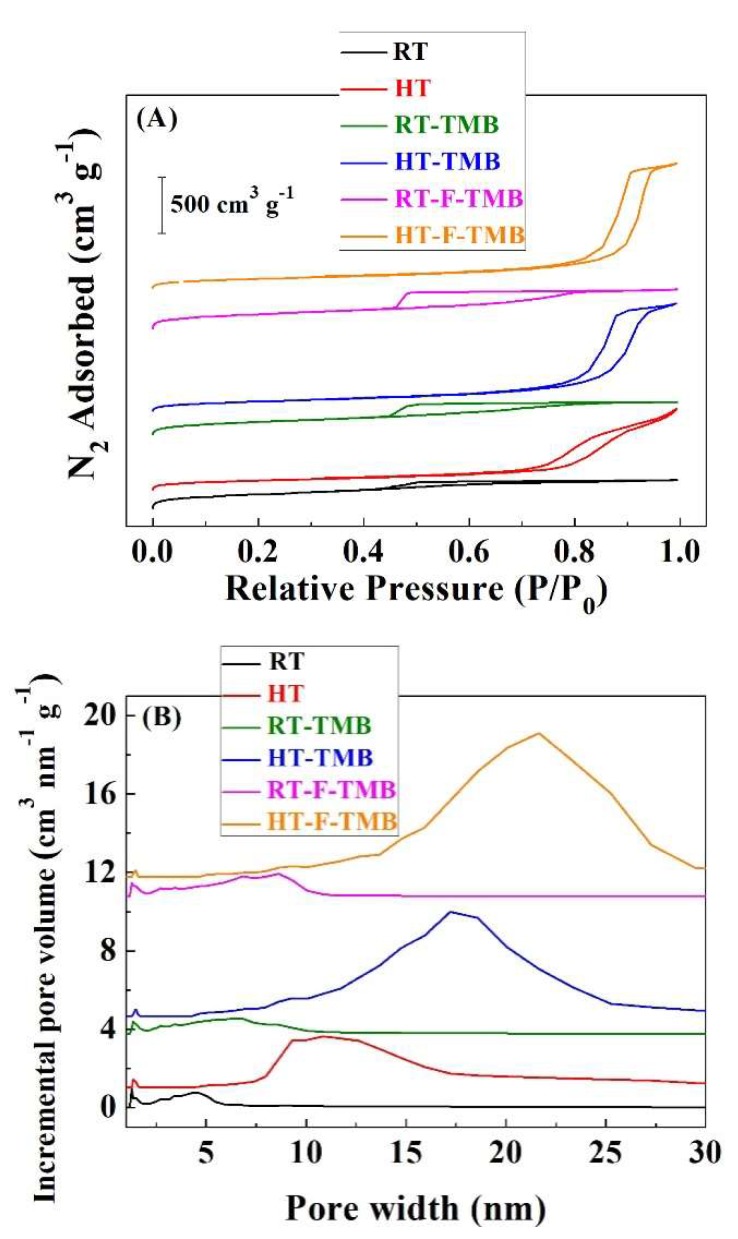
N_2_ adsorption-desorption isotherms (**A**) and pore size distribution (**B**) of the Zr-doped porous silicas.

**Figure 4 ijms-20-00828-f004:**
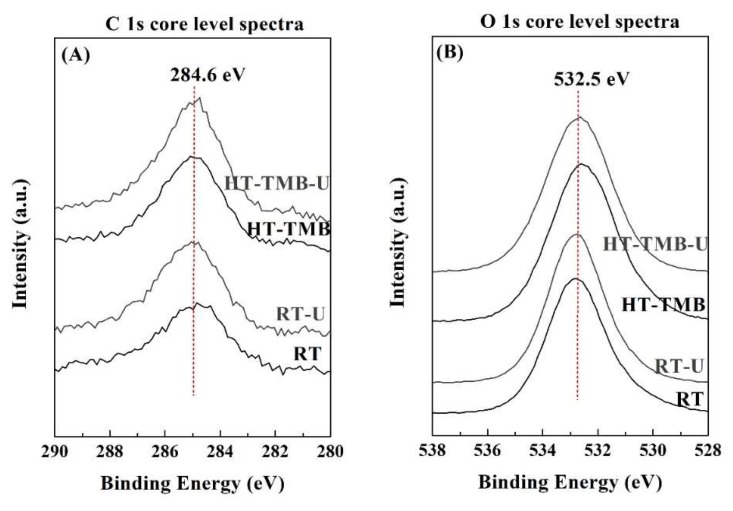
XPS spectra in the C 1*s* (**A**), O 1*s* (**B**), Si 2*p* (**C**), and Zr 3*d* (**D**) regions for the RT and HT-TMB, before and after the catalytic tests.

**Figure 5 ijms-20-00828-f005:**
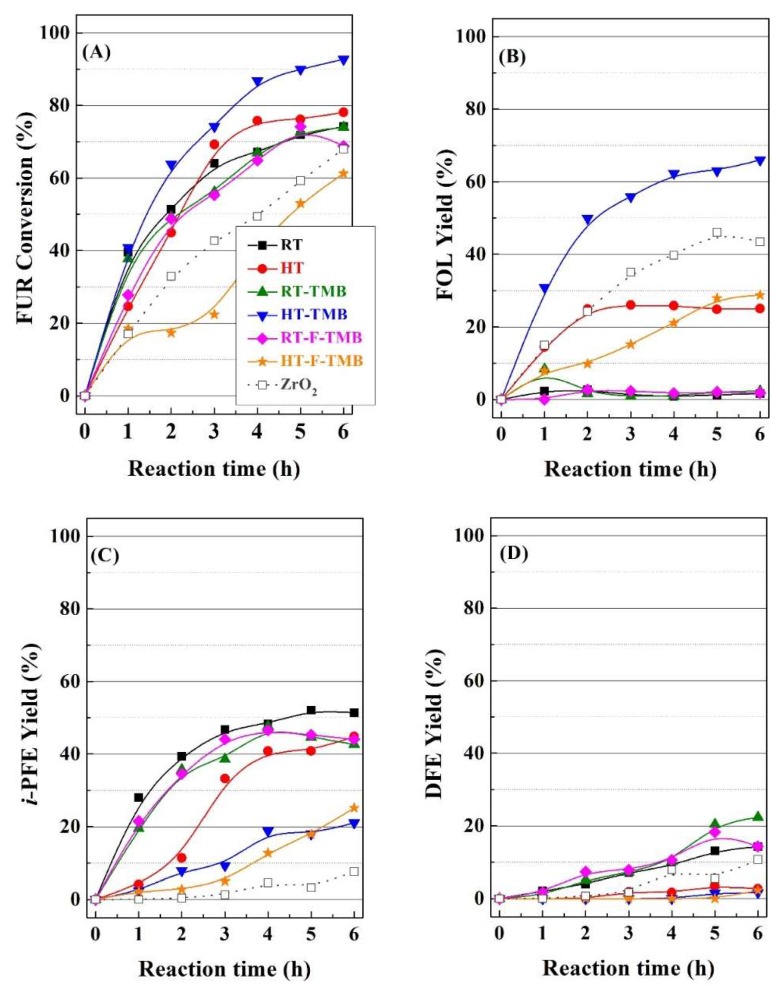
Furfural conversion (**A**), furfuryl alcohol yield (**B**), *i*-propyl furfuryl ether yield (**C**) and difurfuryl ether yield (**D**) in the MPV reaction using Zr-doped porous silica catalysts. (Experimental conditions: 0.1 g of catalyst, reaction temperature: 110 °C, *i*-POH/FUR molar ratio: 50, FUR/catalyst mass ratio: 1).

**Figure 6 ijms-20-00828-f006:**
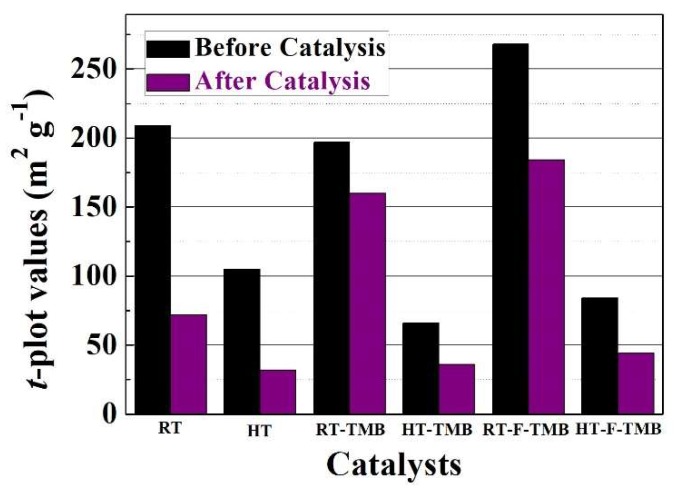
Comparison of the *t*-plot values (m^2^ g^−1^) before and after the MPV reaction. (Experimental conditions: 0.1 g of catalyst, reaction temperature: 110 °C, reaction time: 6 h, *i*-POH/FUR molar ratio: 50, FUR/Catalyst mass ratio: 1).

**Figure 7 ijms-20-00828-f007:**
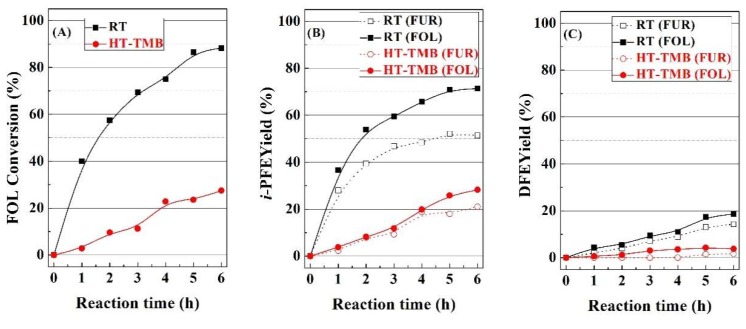
Furfuryl alcohol conversion (**A**), *i*-propyl furfuryl ether yield (**B**) and difurfuryl ether yield (**C**) in the MPV reaction using RT and HT-TMB catalysts. (Experimental conditions: 0.1 g of catalyst, reaction temperature: 110 °C, *i*-POH/FOL molar ratio: 50, FOL/Catalyst mass ratio: 1). (In dashed lines, *i*-PFE and DFE yields using FUR as fed).

**Figure 8 ijms-20-00828-f008:**
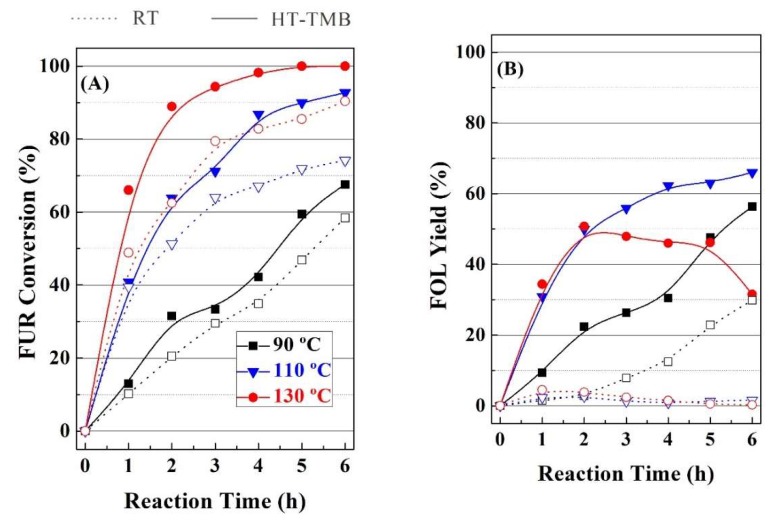
(**A**–**D**) Influence of the temperature on the catalytic performance using RT and HT-TMB catalysts for the MPV reaction of FUR. (Experimental conditions: 0.1 g of catalyst, *i*-POH/FUR molar ratio: 50, FUR/Catalyst mass ratio: 1).

**Figure 9 ijms-20-00828-f009:**
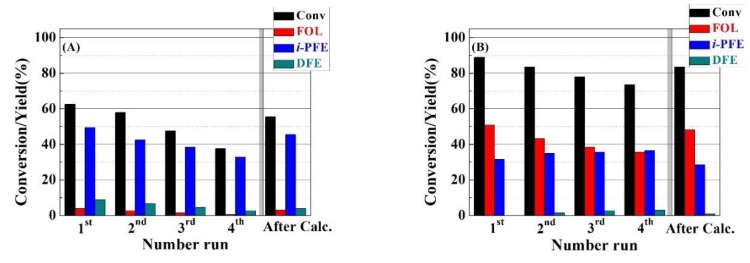
Furfural conversion, furfuryl alcohol yield, *i*-propyl furfuryl ether yield and difurfuryl ether yield for RT (**A**) and HT-TMB (**B**) catalysts. (Experimental conditions: 0.1 g of catalyst, temperature reaction: 130 °C, reaction time: 2 h, run number: 4, *i*-Pr-OH/FUR molar ratio: 50, FUR/Catalyst mass ratio: 1).

**Figure 10 ijms-20-00828-f010:**
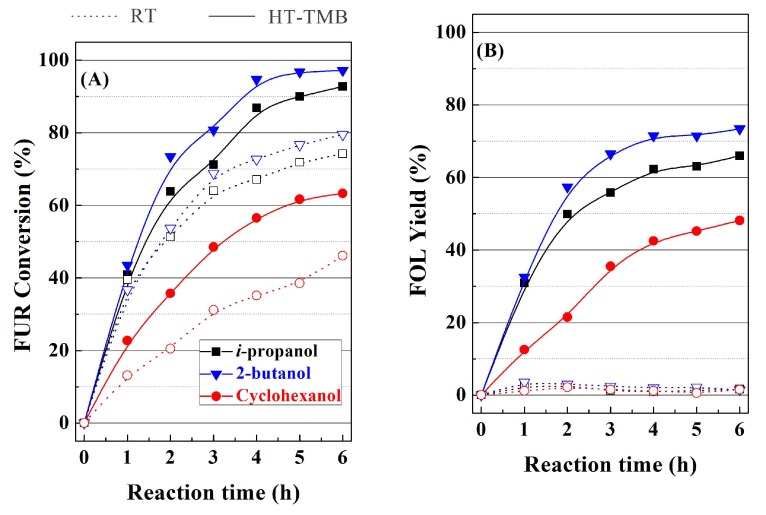
(**A**–**D**) Influence of the sacrificing alcohol on the catalytic performance with RT and HT-TMB catalysts for the MPV reaction of FUR. (Experimental conditions: 0.1 g of catalyst, reaction temperature: 110 °C, *i*-POH/FUR molar ratio: 50, FUR/Catalyst mass ratio: 1).

**Figure 11 ijms-20-00828-f011:**
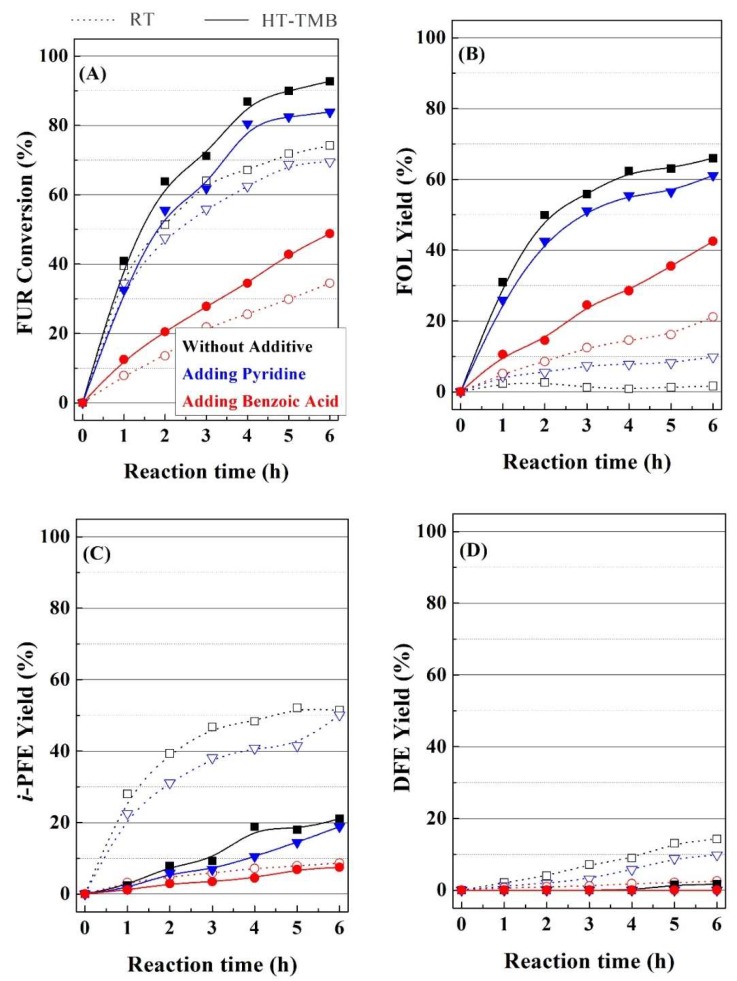
(**A**–**D**) Inhibitory role of the pyridine and benzoic acid in the MPV reaction of FUR using RT and HT-TMB catalysts. (Experimental conditions: 0.1 g of catalyst, reaction temperature: 110 °C, *i*-POH/FUR molar ratio: 50, FUR/Catalyst mass ratio: 1, FUR/Benzoic acid or FUR/Pyridine molar ratio of 1).

**Table 1 ijms-20-00828-t001:** Textural properties of the Zr-doped porous silicas.

Catalyst	S_BET_ (m^2^ g^−1^)	*t*-plot (m^2^ g^−1^)	V_microp_ (cm^3^ g^−1^)	V_p_ (cm^3^ g^−1^)
**RT**	655	209	0.093	0.476
**HT**	390	105	0.044	0.985
**RT-TMB**	638	197	0.087	0.550
**HT-TMB**	410	66	0.024	1.451
**RT-F-TMB**	684	268	0.120	0.627
**HT-F-TMB**	388	84	0.034	1.680

**Table 2 ijms-20-00828-t002:** Atomic concentration deduced by XPS of fresh and spent Zr-dopedsilicas.

Sample	C 1*s*	O 1*s*	Si 2*p*	Zr 3*d*	Si/Zr Molar Ratio
RT	5.24	64.89	28.03	0.79	35.16
HT	4.76	63.26	31.06	0.91	34.14
RT-TMB	5.60	63.44	30.09	0.87	34.58
HT-TMB	4.35	64.38	30.38	0.89	34.13
RT-F-TMB	5.45	63.31	30.46	0.83	36.69
HT-F-TMB	3.89	64.06	31.16	0.89	35.01
RT-U	6.68	62.47	29.95	0.90	31.19
HT-U	7.36	62.29	29.47	0.88	33.48
RT-TMB-U	5.93	63.23	29.95	0.88	34.03
HT-TMB-U	6.14	63.48	28.84	0.98	29.43
RT-F-TMB-U	7.37	61.97	28.48	0.88	32.77
HT-F-TMB-U	4.02	64.02	31.16	0.80	38.45

**Table 3 ijms-20-00828-t003:** List of acronyms of the Zr-doped mesoporous SBA-15 silica used in the MPV reaction of FUR.

Catalyst	Acronym
Zr-doped mesoporous SBA-15 silica synthesized at room temperature	RT
Zr-doped mesoporous SBA-15 silica synthesized under hydrothermal conditions (80 °C)	HT
Zr-doped mesoporous SBA-15 silica with TMB synthesized at room temperature	RT-TMB
Zr-doped mesoporous SBA-15 silica with TMB synthesized under hydrothermal conditions (80 °C)	HT-TMB
Zr-doped mesoporous SBA-15 silica with fluoride and TMB synthesized at room temperature	RT-F-TMB
Zr-doped mesoporous SBA-15 silica with fluoride and TMB synthesized under hydrothermal conditions (80 °C)	HT-F-TMB

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
