# Peer review of "Influence of Structure-modifying Agents in the Synthesis of Zr-doped SBA-15 Silica and Their Use as Catalysts in the Furfural Hydrogenation to Obtain High Value-added Products through the Meerwein-Ponndorf-Verley Reduction"

_ijms, 2019, doi:10.3390/ijms20040828_

Round 1

Reviewer 1 Report

In this study, Zr-doped SBA mesoporous silica was modified with structure-modified agents. The solid was then characterized and used as a catalyst in the furfural hydrogenation via Meerwein-Ponndorf-Verley reduction reaction. Overall, the paper is clearly written and all the points are easy to follow. In my opinion, the works appeared to be performed in a scientifically sound manner and the interpretation of the results was appropriate. Hence, I recommend this manuscript should be published in MMM. Some minor errors:

1.      Abstract: What is FOL?

2.      Line 118: It can also mean that the ZrO2 species is dispersed well on the SBA-15 surface. Please refer to Chem. Eng. J. 243 (2014) 99.

3.      Figure 3: Relative pressure, P/Po, should be used instead of P Po-1 since the former form is used in the text.

4.      Section 4 should be Experimental and not Results.

5.      Line 410 and 420: How big is the volume of the autoclave used for the synthesis? Please check again whether the water molar ratio or chemical composition is correct.  

6.      Many technical errors are detected in the manuscript. E.g. the symbol for degree Celsius, NH3, SN2.

Author Response

In this study, Zr-doped SBA mesoporous silica was modified with structure-modified agents. The solid was then characterized and used as a catalyst in the furfural hydrogenation via Meerwein-Ponndorf-Verley reduction reaction. Overall, the paper is clearly written and all the points are easy to follow. In my opinion, the works appeared to be performed in a scientifically sound manner and the interpretation of the results was appropriate. Hence, I recommend this manuscript should be published in MMM. Some minor errors:

1. Abstract: What is FOL?

FOL is the acronym used for furfuryl alcohol. The authors have incorporated the term furfuryl alcohol in the abstract section to clarify this acronym.

2. Line 118: It can also mean that the ZrO2 species is dispersed well on the SBA-15 surface. Please refer to Chem. Eng. J. 243 (2014) 99.

The authors thank the suggestion of the reviewer. This reference has been incorporated in lines 119-120 as reference [55], together with the next sentence indicating that the Zr species are forming part of the silica walls and/or well dispersed on the internal and external silica surfaces

3. Figure 3: Relative pressure, P/Po, should be used instead of P Po-1 since the former form is used in the text.

The authors again thank this comment. The authors have changed P P0-1 by P/P0 in the x axis of Figure 3.

4. Section 4 should be Experimental and not Results.

The authors have corrected this wrong heading.

5. Line 410 and 420: How big is the volume of the autoclave used for the synthesis? Please check again whether the water molar ratio or chemical composition is correct.

The authors confirm that the molar ratio is right. On the other hand, they have incorporated the total volume of the autoclave in line 420 (90 mL).

6. Many technical errors are detected in the manuscript. E.g. the symbol for degree Celsius, NH3, SN2.

According the suggestion of the reviewer, the technical errors have been corrected along the manuscript.

Reviewer 2 Report

I will recommend to publish this manuscript after major revision.

1) The order of the manuscript is suggested as follow: Introduction, Experimental section, Result and discussion, Conclusion.

2) The authors are suggested to carry out more characterizations to prove the Zr-doped SBA-15

3) The authors are recommended to provide XPS figures. So, the readers will be easy to follow the paper.

4) The authors are suggested to prove the novelty of this manuscript.

Author Response

1) The order of the manuscript is suggested as follow: Introduction, Experimental section, Result and discussion, Conclusion.

The authors agree with this suggestion, but the template provided by the mpdi editorial proposes the next order: (1. Introduction, 2. Results, 3. Discussion, 4. Materials and Methods, 5. Conclusions), as found in the next link:

https://www.mdpi.com/journal/ijms/instructions#submission

2) The authors are suggested to carry out more characterizations to prove the Zr-doped SBA-15.

The authors thank the suggestion of the reviewer. In the present work, different Zr-doped mesoporous silica materials have been synthesized, whose structural ordering was elucidated by XRD, N2 adsorption-desorption at -196ºC and TEM. The incorporation and dispersion of Zr species were evaluated by XPS, whereas the generation of both acid and basic sites associated to Zr species has been proved by NH3-TPD and CO2-TPD, respectively.

3) The authors are recommended to provide XPS figures. So, the readers will be easy to follow the paper.

Following the advice of the reviewer, the authors have incorporated the X-ray photoelectron spectra of the RT and HT-TMB catalysts, before and after the catalytic process, as representative examples of catalysts prepared in the present work.

Figure 4. XPS spectra in the C 1s (A), O 1s (B), Si 2p (C) and Zr 3d (D) regions for the RT and HT-TMB before and after the catalytic tests.

4) The authors are suggested to prove the novelty of this manuscript.

The authors indicated in the introduction section the aim of the present work (lines 103-104) that summarizes the novelty of this manuscript

The present research evaluates the influence of the width and length of pores on the catalytic behavior in the MPV reaction of FUR to obtain valuable products.”

On the other hand, the conclusion section also shows how the modification of the textural parameter is a key factor in the catalytic activity and the selectivity pattern (lines, 487-497)

Zr-doped porous silicas were evaluated in the catalytic transfer hydrogenation of FUR with alcohols. The catalytic results revealed that the textural properties are directly related to the FUR conversion, reaching the highest activity for the HT-TMB catalysts. In the same way, the selectivity pattern is also highly influenced by the pore size distribution and the size of channels. Thus, the catalyst synthesized under hydrothermal conditions, with a larger pore size, tends to form FOL as main product through the MPV reaction. However, the Zr-doped silicas synthesized at room temperature, that display narrower pores, form i-PFE and DFE as products, with FOL in lower proportions. These data suggest that, in a first step, FOL is formed by the MPV reaction, and then FOL is etherified with another FOL molecule, or with the sacrificing alcohol present in the reaction medium, on acid sites”.

Round 2

Reviewer 2 Report

I accept the author answers!